


# Recent trends and variability in river discharge across northern Canada

Stephen J. Déry[1], Tricia A. Stadnyk[2], Matthew K. MacDonald[1,2], Bunu Gauli-Sharma[1]

[1]Environmental Science and Engineering Program, University of Northern British Columbia, Prince
George, British Columbia, Canada
[2]Department of Civil Engineering, University of Manitoba, Winnipeg, Manitoba, Canada

*Correspondence to*: Stephen J. Déry (sdery@unbc.ca)

**Abstract.** This study presents an analysis of the observed interannual variability and interdecadal trends
in river discharge across northern Canada for 1964–2013. The 42 rivers chosen for this study span a
combined gauged area of $5.26 \times 10^6$ km$^2$ and are selected based on data availability and quality, gauged
area, and record length. Interannual variability in river discharge is greatest for the Eastern Arctic Ocean
(coefficient of variation CV = 16%) due to the Caniapiscau River diversion into the La Grande Rivière
system for enhanced hydropower production. Variability is lowest for the study area as a whole (CV =
7%). Based on the Mann-Kendall Test (MKT), no significant ($p > 0.05$) trend in annual discharge from
1964-2013 is observed to the Bering Sea, Western Arctic Ocean, Western Hudson and James Bay, and
Labrador Sea; for northern Canada as a whole, however, a statistically-significant ($p < 0.05$) decline of
102.8 km$^3$ (25 year)$^{-1}$ in discharge occurs over the first half of the study period followed by a
statistically-significant ($p < 0.05$) increase of 208.8 km$^3$ (25 year)$^{-1}$ in the latter half period. Increasing
(decreasing) trends in river discharge to Eastern Hudson and James Bay (Eastern Arctic Ocean) are
largely explained by the Caniapiscau diversion to the La Grande Rivière system. Strong regional
variations in seasonal trends of river discharge are observed, with overall winter (summer) flows
increasing (decreasing, with the exception of the most recent decade) partly due to flow regulation and

storage for enhanced hydropower production along Hudson and James Bay, the Eastern Arctic Ocean and Labrador Sea. Flow regulation also suppresses the natural variability of river discharge, particularly during cold seasons.

**Keywords:** river discharge, northern Canada, climate change, flow regulation, Arctic Ocean

## 1 Introduction

The pan-Arctic region is experiencing the highest rates of warming on Earth, substantially altering its environment and ecosystems (Serreze et al., 2000; Hinzman et al., 2005; Callaghan et al., 2011). As a result of this warming, significant declines in Arctic sea ice and pan-Arctic snow cover extent are being

observed, inducing a positive snow/ice-albedo feedback on warming (Serreze et al., 2007; Déry and Brown, 2007; Shi et al., 2011; Hernández-Henríquez et al., 2015). In turn, reductions in Arctic sea ice and pan-Arctic snow cover affect atmospheric circulation, with evidence emerging for a stronger meridional (rather than zonal) pattern in the Northern Hemisphere during recent years (Liu et al., 2012; Francis and Vavrus, 2012). The pan-Arctic hydrological cycle is also showing signs of change, as

warmer conditions enable enhanced moisture transport into the pan-Arctic with concomitant increases in precipitation (Zhang et al., 2013) and intensification of the land surface hydrological cycle (Rawlins et al., 2010; Déry et al., 2009). The result has been increasing river discharge in the six principal rivers of Eurasia draining into the Arctic Ocean (Peterson et al., 2002; McClelland et al., 2004).

Findings from Peterson et al. (2002) for the Eurasian continent led many researchers to inquire whether similar trends were being observed in North America. Motivated by this, Déry et al. (2005a) conducted

a comprehensive analysis of river discharge into Hudson, James and Ungava Bays spanning 1964 to 2000. In contrast to the findings of Peterson et al. (2002), Déry et al. (2005a) reported a recent 13% decline in river discharge to Hudson, James, and Ungava Bays. Déry and Wood (2005) expanded this effort to examine streamflow trends in 64 rivers draining all of northern Canada with the exception of

the Canadian Arctic Archipelago (CAA). Over the period 1964-2003, Déry and Wood (2005) found a 10% decline in discharge for rivers draining northern Canada, consistent with recent decreases in precipitation over the study area. This unexpected result was attributed partly to the strong relationship between the Arctic Oscillation and river discharge in northeastern Canada (Déry and Wood, 2004). Following this, McClelland et al. (2006) assembled data from both Eurasia and North America to

provide a complete pan-Arctic view on recent trends in river discharge. Using a consistent study period and method for trend analysis, they concluded that pan-Arctic river discharge increased by 5.6 km$^3$ yr$^{-1}$ yr$^{-1}$ from 1964 to 2000, despite observed declines in river discharge to Hudson, James and Ungava Bays.

A decade has now passed since the work of Déry et al. (2005a), Déry and Wood (2004, 2005) and McClelland et al. (2006), offering the opportunity to reassess trends in river discharge across northern Canada, and to evaluate if trends are now more aligned with those observed in Eurasia. This study therefore investigates trends and variability in river discharge for 42 principal rivers draining northern Canada over a 50-year period (1964-2013). The research question motivating this effort is whether or

not river discharge in northern Canada shows a continued decrease in the twenty-first century as first reported by Déry and Wood (2005). The effects of flow regulation and climate variability are both



considered in our analyses, with emphasis on the interdecadal seasonal variability in river discharge. Then the discussion provides a comparison with previous studies, a review of anthropogenic effects on observed trends and variability in river discharge across northern Canada, and an overview of the potential physical impacts to the marine environment. A summary of the study's main findings and avenues for future work closes the paper.

## 2 Study Area

### 2.1 Physical Setting and Climate

A vast portion of Canada and parts of the northern United States drain northward to the Bering Strait, Arctic Ocean, Hudson and James Bays, Hudson Strait and Labrador Sea (Fig. 1). The CAA also drains into the Arctic Ocean but remains largely ungauged (Spence and Burke, 2008), with only two small rivers in this study (see Sect. 3.1). Six separate drainage basins are considered here (from west to east): Bering Strait, Western Arctic Ocean, Western Hudson and James Bay, Eastern Hudson and James Bay, Eastern Arctic Ocean, and Labrador Sea. The gauged area totals $5.26 \times 10^6$ km$^2$, more than half of the Canadian land surface area (Table 1). The Canadian provinces of British Columbia (BC), Alberta, Saskatchewan, Manitoba, Ontario, Québec and Newfoundland/Labrador along with the Yukon, Northwest and Nunavut Territories form part of the study area. Some tributaries of the Nelson River drain a small portion of the north-central United States, namely in Montana, North and South Dakota, and Minnesota. Among the larger systems are the Yukon, Mackenzie, Back, Thelon-Kazan (collectively referred to as Chesterfield Inlet hereinafter), Churchill (Manitoba), Nelson, Hayes (Manitoba), Albany, Moose, La Grande, Koksoak and Churchill (Labrador) Rivers (Table 1).

The vegetation and land cover varies markedly across the vast area drained by northern Canada's rivers. The northern Rocky Mountains with peaks approaching 4000 m above sea level in the headwaters of the Yukon, Mackenzie and Nelson Rivers have bare rocks, glaciers and snow with limited vegetation such as lichens and mosses. Grasslands of the central Canadian Prairies and American northern Great Plains

subject to intense agricultural activity cover the central portion of the Nelson River Basin. Further north and to the east, boreal and taiga forests of the Canadian Shield span a vast portion of the study area. Arctic tundra underlain by permafrost covers the northernmost portions of these drainage basins. Several large bodies of water including Great Bear, Great Slave and Reindeer Lakes, along with Lakes Athabasca, Manitoba, Winnipegosis and Winnipeg and countless smaller lakes, ponds and wetlands

form natural reservoirs in this system. Large artificial reservoirs developed for hydropower production exist in the study area as well, most prominently in the La Grande Rivière, Nelson, and Churchill (both in Manitoba and Newfoundland/Labrador) River Basins (see Sect. 2.2).

The climate also varies substantially across the study area. In mountainous terrain of northwestern

Canada, mean annual air temperatures remain below 0°C with abundant snowfall dominating the form of precipitation. The Canadian Prairies and northern American Great Plains to the lee of the western Cordillera are relatively warm and dry (mean annual total precipitation of 300-500 mm), with most of the precipitation occurring during summer. The boreal and taiga forests experience relatively cool and wet climate regimes (mean annual total precipitation of 500-1000 mm), with both abundant rainfall and

snowfall. On the Arctic tundra, cold temperatures (mean annual air temperature < -10°C) and snowfall dominate the climate. The seasonal snow cover typically lasts 4-6 months on the Canadian Prairies, ~6

months in the boreal forest, and 6-8 months in Arctic tundra and mountainous terrain (McKay and Gray, 1981). Given these climate regimes, most unregulated rivers of northern Canada exhibit a nival regime, with low flows in winter when water is stored in the seasonal snowpack, then high flows during the snowmelt-driven freshet in spring and early summer (Déry et al., 2005a). A summer recession driven by high evapotranspiration rates follows, with possible secondary peak flows in autumn caused by the frequent passage of synoptic storms (Déry et al., 2005a). High flows at times occur in summer as well in small creeks and rivers associated with severe convective activity or at larger scales when associated with intense synoptic storms. In contrast, some regulated systems exhibit low temporal variability in flows with daily fluctuations arising from hydropower demand and generation (Woo et al., 2008; Déry et al., 2011). In areas affected by permafrost, hydrological responses are relatively rapid given the limited infiltration capacity of frozen soils (Woo, 1986). Glaciers in the northern Rocky Mountains and other mountain chains also supply additional meltwater in late summer and early fall, particularly during warm, dry years (Marshall et al., 2011).

## 2.2 Regulated Systems

Several rivers in the study area are regulated for hydropower production, but also for flood protection, irrigation, industrial and recreational purposes; and are thus considered moderately to strongly fragmented (Dynesius and Nillson, 1994). The highly fragmented Nelson River Basin has a long history of hydropower development, with hydroelectric generation beginning in 1906 on the Pinawa Channel of the Winnipeg River System (Manitoba Hydro, 1998). Since then, there has been a proliferation of dams constructed along the Nelson River's main stem and several of its tributaries. Reservoirs such as the

artificial Lake Diefenbaker (formerly a section of the South Saskatchewan River) and natural Lake

Winnipeg and Southern Indian Lake allow seasonal water storage in this system that is managed

depending on inflows, hydropower demand, flood protection and governmental regulations. In 1976,

Manitoba's Churchill River was partially diverted through the Rat and Burntwood Rivers (with water

5   releases controlled at the Notigi Control Structure) for enhanced hydropower production on the lower

Nelson River. An additional capacity of 7 km$^3$ of water storage in Southern Indian Lake was developed

in the process as it thereafter became managed (Déry and Wood, 2005). Since then, approximately 75%

of the annual flows in Manitoba's Churchill River are diverted into the lower Nelson River, greatly

diminishing the Churchill River's annual inflows to Hudson Bay (Newbury et al., 1984).

Another highly fragmented system is La Grande Rivière where the massive James Bay hydroelectric

complex has been developed since the mid-1970s by Hydro-Québec (Hernández-Henríquez et al.,

2010). As a result of this, several large reservoirs with a storage capacity now approaching 200 km$^3$

have been built and are managed depending on hydropower demand and consumption. Development of

15   the James Bay hydroelectric complex has diverted portions of the Eastmain and Opinaca Rivers starting

in 1980, the upper Caniapiscau River (a major tributary of the Koksoak River) in 1982, and the Rupert

River in 2009 to La Grande Rivière's basin (Déry et al., 2005a). Of note, the Caniapiscau River

diversion (area = 36,900 km$^2$) induces an inter-basin transfer of 45% of its flows or 748 m$^3$ s$^{-1}$ from the

Eastern Arctic Ocean toward the Eastern Hudson and James Bay system (Roy and Messier, 1989). The

20   overall drainage basin area for La Grande Rivière has now effectively doubled in size to surpass

200,000 km$^2$ (Roy and Messier, 1989; Hydro-Québec, 2008). Just to the east of the Caniapiscau

Reservoir lies Newfoundland and Labrador's Churchill River that is also managed for hydropower production. Construction of hydroelectric facilities at Churchill Falls began in 1967 and they have been fully operational since 1974. This has led to the creation of the Smallwood Reservoir with water storage capacity of 33 km$^3$ (Déry and Wood, 2005). In Ontario, the Moose River and its many tributaries are

5 highly fragmented by a series of 40 hydroelectric dams with development beginning in 1911 (Benke and Cushing, 2005). However, these are mainly run-of-river projects with little storage capacity, exerting less influence on downstream flows. While the Mackenzie River's main stem is unregulated, one of its major tributaries, the Peace River (basin area ~293,000 km$^2$), remains managed for hydropower production. Construction of the W. A. C. Bennett Dam from 1968 to 1972 created the

10 Williston Reservoir with a storage capacity of 70 km$^3$. This has led to an attenuation of the seasonal cycle in downstream flows, affecting the recharge of the Peace-Athabasca Delta (Rasouli et al., 2013). Other rivers moderately affected by fragmentation in northern Canada include the Grande Rivière de la Baleine, Nottaway and Albany Rivers (Dynesius and Nillson, 1994).

## 3 Data and Methods

### 3.1 Data and Study Period

This study examines 42 main rivers of northern Canada for which daily hydrometric data from gauging stations are available (Table 1). The principal source of the hydrometric data remains the Water Survey of Canada (http://www.ec.gc.ca/rhc-wsc/), with supplemental data from the Direction d'Expertise

Hydrique du Québec (http://www.cehq.gouv.qc.ca/) for rivers in that province from 2000 to 2013. Manitoba Hydro and Hydro-Québec also provide daily hydrometric data for the regulated Nelson River and La Grande Rivière, respectively. Gauges furthest downstream on a river's main stem are chosen to

obtain the maximum spatial coverage and most accurate estimates of total inflows to the coastal ocean. Additional criteria used for the selection of the 42 rivers are: 1) >30 years of data availability over 1964-2013 (the study period); 2) gauged area > 1000 km$^2$; and 3) outlets to the coastal ocean in northern Canada. Note that only the Canadian portions of the Yukon River and its tributary the Porcupine River

are included here, although additional hydrometric data are available for the former near its outlet to the Bering Strait (e.g., Walwoord and Striegl, 2007). This is to establish the direct contribution of Canadian rivers to discharge into the coastal ocean. Apart from the 42 rivers selected for this study, additional hydrometric data for tributaries that flow downstream from a gauge on a river's main stem are also included in the development of the discharge time series (Table 1). In that case, results are presented

collectively and the systems are then referred to by the river's main stem. For example, results for the Peel River are added to the Mackenzie River (at Arctic Red River) as their hydrometric gauges are upstream of the confluence of these two rivers. Section 3.2.1 provides details of the construction of the river discharge time series when such situations arise.

While discharge measurements remain highly constrained observational data, errors arise nonetheless

during the collection process (Lammers et al., 2001; Shiklomanov et al., 2006). Sources for these errors range from the collection method, sampling frequency, environmental conditions (e.g., under ice cover, backwater effects during ice jams, flood events, beaver dams and vegetation) and the local geography (presence or absence of a flood plain). Errors in measurements typically range from ±2-5% in the absence of both a flood plain and an ice cover (Lammers et al., 2001); however, errors in measurement

increase to ±5-12% in the presence of either a flood plain and/or an ice cover. Errors may reach or even exceed these values during peak and low flows as well (Pelletier, 1988; Di Baldassarre and Montanari,

2009). While a comprehensive analysis of errors in discharge measurements is beyond the scope of this work, it is assumed this study's observational data are subject to similar errors reported by Pelletier (1988), Lammers et al. (2001), Shiklomanov et al. (2006) and Di Baldassarre and Montanari (2009). Caution is also needed in interpreting results for the Thelon-Kazan Rivers (Chesterfield Inlet) as a

change in recording methodology in the mid-1980s may lead to spurious trends in that system (Déry et al., 2011). Finally, flow measurement error likely decreases over time in this study as sampling methods become more reliable and increasingly more automated.

The study period covers 50 years, starting in 1964 and ending in 2013. While long term hydrological records are necessary to distinguish the impacts of decadal climate variability from climate change on

streamflow, northern Canada has a paucity of hydrometric data prior to 1964 (Mlynowski et al., 2011). The rapid expansion of northern Canada's hydrometric network in the mid-1960s, particularly on main stem rivers with gauging stations installed near their outlets, allows the study period here to begin in 1964. There are relatively long term (century scale) hydrometric data in more southern tributaries of some systems, however, including the Mackenzie, Nelson and Moose Rivers. Availability of long term

hydrometric data for rivers draining the CAA remains limited to Freshwater Creek near Cambridge Bay on Victoria Island (gauged area 1490 km$^2$ draining into the Western Arctic Ocean) and the Sylvia Grinnell River near Iqaluit on Baffin Island (gauged area 2980 km$^2$ draining into the Eastern Arctic Ocean). Thus only 0.3% of the CAA has available hydrometric data, implying the results are not representative of this vast region where glaciers and ice caps are in rapid retreat (Gardner et al., 2011).

In fact, most of northern Canada falls well below the World Meteorological Organization's (WMO's) standards for hydrometric gauge density, imposing limitations on this effort (Coulibaly et al., 2013).

More recent hydrometric data (post 2013) remain largely unavailable due to ongoing quality control processes by various governmental agencies; thus only operational (historical) data from the Water Survey of Canada and its provincial/territorial partners are used in this study since provisional (near real-time) data posted online have not yet undergone quality control and analysis.

## 3.2 Methods

### 3.2.1 Time Series Construction

Following quality control and analysis, daily streamflow data (in $m^3 \ s^{-1}$) are compiled and transformed to seasonal and annual time series of discharge (in $km^3 \ yr^{-1}$) for 42 rivers in northern Canada (Table 1).

The four seasons are taken here as winter (January to March), spring (April to June), summer (July to September) and fall (October to December). This selection is somewhat arbitrary since the actual duration of each season varies greatly from region to region (e.g., wintertime conditions can easily persist for six or more months on the Arctic tundra). For some systems (most notably the Mackenzie, Nelson, and Moose Rivers), hydrometric data from tributaries downstream of main stem gauging

stations are included in the database (such as the Peel River with the Mackenzie River). Data from these tributaries are then added to the concurrent time series for the river's main stem and are referred to simply by the principal waterway. Streamflow data for regulated or partially diverted rivers are not naturalized in this study. Motivation for this strategy lies in the study's main objective of quantifying actual discharge to the coastal ocean, irrespective of the effects of climate change, land use and land

cover change, and flow regulation. Likewise, discharge data are not adjusted to account for the filling of

large reservoirs such as in the La Grande Rivière, Nelson and Mackenzie (Peace River) systems, leading

to a better understanding of the impacts of changing river discharge in northern Canada.

Construction of the discharge time series when gaps exist follows a two-step process (as needed) similar

to Déry et al. (2005a). First, daily hydrometric data from the gauging station furthest downstream and

near a river's outlet to the coastal ocean are used to represent the watershed. If unavailable, then an

upstream gauge is used and streamflow data are adjusted to account for the missing contributing area

(Déry et al., 2005a). In several instances, this includes combining data from two or more tributaries

upstream from a main stem river's gauge (e.g., the Waswanipi and Bell Rivers for the Nottaway River

after 1982). When upstream gauges remain unavailable, a secondary step is taken to fill in data gaps.

Here a daily climatology of streamflow (or mean annual hydrograph) is constructed based on the

availability of data over the period of record. Missing data on a given day are then in-filled with the

daily mean value of streamflow over the available period of record. For Manitoba's Churchill River and

Québec's Eastmain, Caniapiscau and Rupert Rivers, separate climatologies of daily streamflow are

constructed for the periods prior to and after flow diversions (see Sect. 2.2). This is a more appropriate

gap filling strategy for these rivers prior to and subsequent to diverted flows. The impacts of this gap

filling strategy on discharge trend and statistical analyses are discussed in Sect. 3.2.2.

There are substantial gaps in some of the discharge time series that are in-filled. Most notable are gaps

in the first few years of the study period as the network of hydrometric gauges was being enhanced,

particularly in remote rivers of northern Canada. Between 1970 and 1990, the gauged area in rivers of

northern Canada stabilized until some notable reductions in northern Ontario in the mid- to late 1990s
and in northern Québec in the early to mid-2000s (Mlynowski et al., 2011). Decreases in gauged area
persisted into the late 2000s, with a steady recovery since then (Coulibaly et al., 2013). The most
prominent data gaps in northern Ontario are in the Ekwan River (1964-1966 and 1996-2010), the

Severn River (1995-2006), the Albany, Winisk and Attawapiskat Rivers (1996-1998), and the main
stem Moose River (1998-2001). In northern Québec, pronounced gaps exist for all rivers draining into
the Eastern Arctic Ocean, primarily between 2000 and 2008. Furthermore, data downstream of the
diverted flows of the Eastmain and Rupert Rivers are lacking after 2005 and in-filled with estimates of
mean daily flow accounting for their partial diversions to La Grande Rivière. Hydrometric data for

some smaller systems (e.g., Freshwater Creek, and the Firth, Ellice and Sylvia Grinnell Rivers) in
Canada's northern territories are often only seasonally available. In absence of wintertime hydrometric
data, daily discharge is assumed to be zero as these rivers likely freeze to their beds (e.g., Woo, 1986).
Following these steps, time series are aggregated to six regional drainage basins based on the bodies of
water they drain into: the Bering Strait (Canadian portion only), Western Arctic Ocean, Western

Hudson and James Bay, Eastern Hudson and James Bay, Eastern Arctic Ocean (Ungava Bay/Hudson
Strait), and Labrador Sea (see Fig. 1).

### 3.2.2 Statistical and Trend Analyses

Statistics of the mean, standard deviation (SD) and coefficient of variation (CV = SD/mean) in annual

and seasonal river discharge for each of the six drainage basins and total gauged area are first computed.
Linear trend analysis follows the approach of Déry et al. (2005a, 2011) by employing the Mann-Kendall
Test (MKT; Mann, 1945; Kendall, 1975). The Sen's slope estimator provides the magnitude of the trend

while a probability value ($p$-value) of 0.05 quantifies statistically-significant trends in this work. Prior to trend analysis, time series of annual and seasonal river discharge are tested for serial correlation. If the lag-1 auto-regression for either annual or seasonal time series of river discharge attains $p < 0.05$, then 'pre-whitening' of the data following Yue et al. (2002) is performed. Seasonal autocorrelations in total

5 annual discharge across northern Canada are also presented. Both temporal analyses for the six regions and all of northern Canada and spatial analyses for each of the 42 rivers are presented.

Gap-filling can influence the magnitude of MKT trends. Missing data replaced by climatological values reduces the variability (both the SD and CV) in discharge, attenuating linear trends. While overall

annual and seasonal discharge statistics are assessed only from the available records, care must be used in interpreting linear trends, particularly in systems where large gaps arise (see Sect. 3.2.1). Other rivers exhibit strong trends that appear from inter-basin diversions, which must also be interpreted in the appropriate context.

Additional analyses on seasonal variability of river discharge for each decade (1964-1973, 1974-1983, 1984-1993, 1994-2003, and 2004-2013) are performed for eight regulated (R) and seven 'matching' unregulated (U) rivers. Rivers chosen for this comparison are the Nelson/Churchill in Manitoba (R) with the Seal and Hayes rivers (U), the Moose (R) and Albany (U) rivers, the Rupert and Eastmain rivers (R) with the Nottaway (U) River, La Grande Rivière (R) and Grande Rivière de la Baleine (U),

the Koksoak (through its tributary the Caniapiscau, R) and à la Baleine (U) rivers, and the Churchill River in Labrador (R) and Eagle River (U). The 'matching' unregulated systems are selected for their





proximity to corresponding regulated systems, similar climatic and hydrological regimes, and comparable physiography and drainage areas. Box and whisker plots showing the median, interquantile ranges and the 5th and 95th percentiles in the CV of seasonal river discharge per decade are contrasted for regulated and matching unregulated systems. For proper interpretation of this interdecadal analysis,

it is important to review the timeline of hydroelectric infrastructure development in northern Canada. The majority of hydroelectric development occurred during the 1974-1983 decade, focusing on the construction of dams and diversions in the Nelson River and La Grande Rivière systems. The 1964-1973 period denotes the pre-regulation period in this study (although the Nelson and Moose Rivers were fragmented prior to 1964); 1974-1983 represents the construction period when large dams and

diversions on the Churchill, Eastmain, La Grande Rivière and Koksoak (Caniapiscau) systems were introduced, and development of the Nelson River continued. Therefore 1984-2013 marks the post-regulation period (however, regulation [diversion] of the Rupert River did not commence until the 2004-2013 decade).

## 4 Results

### 4.1 Temporal Analyses

Table 1 lists comprehensive statistics and trend analyses for this study's 42 rivers over 50 years. Mean annual discharge is highest in the Mackenzie (311.4 km$^3$ yr$^{-1}$), Nelson (102.7 km$^3$ yr$^{-1}$), La Grande Rivière (84.2 km$^3$ yr$^{-1}$) and Yukon (77.3 km$^3$ yr$^{-1}$) rivers. With the recent diversion of the Rupert River,

mean annual discharge in La Grande Rivière regularly exceeds 100 km$^3$ yr$^{-1}$, with a record 129.2 km$^3$ yr$^{-1}$ in 2013. Variability (expressed by the CV) in annual discharge remains relatively low (high) in the large (small) basins. The CV in annual discharge remains relatively high in unregulated rivers of the

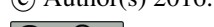



Western Arctic Ocean and Western Hudson and James Bay with a maximum of 70% for Manitoba's
Churchill River where diverted and regulated flows enhance year-to-year variability. Few statistically-
significant trends arise for the study period, with the notable exception of rivers affected by diversions
and flow regulation.

Table 2 provides aggregated statistics of the mean, SD, CV and trend of annual discharge for six regions
of northern Canada from 1964-2013. Mean annual discharge ranges from 77.0 km$^3$ yr$^{-1}$ in the Labrador
Sea to 349.9 km$^3$ yr$^{-1}$ in the Western Arctic Ocean, with a total of 1154.1 km$^3$ yr$^{-1}$ for the gauged area of
northern Canada. Considerable interannual variability in discharge exists, with the CV ranging spatially
from 9% to 16%, although this value diminishes to 7% for the system as a whole. Relatively constant
discharge to the Eastern Arctic Ocean in the 2000s arises from large data gaps in this region and the in-
filling strategy used in the present study (Fig. 2). There is no significant trend in the 1964-2013 annual
discharge to the Bering Strait, Western Arctic Ocean, Western Hudson and James Bay and Labrador
Sea. Nonetheless, a 5-year running mean applied to the discharge time series shows rising annual
discharge since 1990 for Western Hudson and James Bay. High flows in the late 2000s including record
high annual river discharge (438.8 km$^3$ yr$^{-1}$) to Western Hudson and James Bay in 2005, follows near-
record low annual amounts (263.1 km$^3$ yr$^{-1}$) in 2003. Similarly, a reversal from record low river
discharge (258.5 km$^3$ yr$^{-1}$ in 1995) precedes record high river discharge (419.0 km$^3$ yr$^{-1}$ in 1997) to the
Western Arctic Ocean. Persistent low annual discharge to the Eastern Arctic Ocean from 1982 onward
arises largely from the inter-basin diversion of the Caniapiscau River to La Grande Rivière, enhancing
discharge in Eastern Hudson and James Bay. Discharge to the Labrador Sea shows strong decadal



fluctuations that may be associated with climate variability such as different phases of the Arctic Oscillation. For northern Canada as a whole, a modest (but insignificant) positive trend of 0.21 km$^3$ yr$^{-1}$ yr$^{-1}$ is found, equivalent to a change of <1% in mean annual discharge (Fig. 3). The 5-year running mean shows at least two distinct phases: a declining trend in the first half of the study period followed

by increasing discharge until the early 2010s. Indeed, MKT analyses reveal a significant decline of 102.8 km$^3$ (25 year)$^{-1}$ for 1964-1988 followed by a significant increase of 208.8 km$^3$ (25 year)$^{-1}$ for 1989-2013. A case for a possible third phase with relatively stable river discharge across northern Canada could also be argued for the central portion of the record (1985-1995).

Seasonally, spring and summer river discharge to Eastern Hudson and James Bay and the Labrador Sea decline as flows during those seasons are retained in reservoirs and released in winter for hydropower production when demand peaks (Fig. 4). The strong seasonality in flows observed in the 1960s and early 1970s in these two regions nearly vanishes in the 2000s, most notably in the Eastern Hudson and James Bay region. For the mostly unregulated Bering Strait and Western Arctic Ocean drainage basins,

strong seasonality in flows persists through the 50-year study period. There is also a modest, statistically-significant increase in winter flows to Western Hudson and James Bay and marked declines in spring flows to the Eastern Arctic Ocean. Furthermore, high summer flows to the Bering Sea and to Western Hudson and James Bay arise in the 2000s. Changes in seasonality to Western Hudson and James Bay remain less pronounced than those in Eastern Hudson and James Bay owing to the dominant

type of regulation (run-of-river) in the Nelson River versus the large storage capacity of reservoirs in the La Grande Rivière system. For the system as a whole, spring and summer flows are nearly equal

(~390 km$^3$ yr$^{-1}$), given the large (natural and artificial) storage capacity, vast areas and high latitudes of the study basins that delay (into summer) the release of snow meltwater to the Arctic Ocean and adjacent northern seas. A significant decline in summer flows, however, appears over the study period, with perhaps the exception of the 1994-2013 decade (Fig. 5). This is compensated by a gradual and significant increase in winter flows as water releases from reservoirs for hydropower production augments in post-construction decades. In contrast, autumn river discharge shows no trend between 1964 and 2013. Cross correlations between time series of seasonal discharge shows that spring/summer ($r = 0.41$, $p < 0.05$), summer/fall ($r = 0.60$, $p < 0.05$) and fall/winter of the following calendar year ($r = 0.31$, $p < 0.05$) are temporally correlated, showing persistence in seasonal flows. Despite this, there are no statistically-significant correlations between time series of river discharge in other seasons, including between spring and fall of a given year.

## 4.2 Spatial Analyses

Large, significant trends in the 1964-2013 annual river discharge occur mainly in regulated systems (Fig. 6). A notable exception is the Chesterfield Inlet that shows a 4.9 km$^3$ (50 year)$^{-1}$ increase across the study period, noting however the potential recording issues for this system (see Sect. 3.1). Rivers draining to the Eastern Arctic Ocean and Labrador Sea nearly all show declines between 1964 and 2013, in part owing to flow regulation, retention and diversions; nearby unregulated systems also show declines. Couplets of large positive and negative trends to Western and Eastern Hudson and James Bay arise from diverted flows from one system to another. Otherwise, there are no significant trends in river discharge to the Bering Strait and Western Arctic Ocean during the study period. Seasonal analyses



reveal a consistent pattern toward greater winter discharge across northern Canada (with a few exceptions) in both regulated and unregulated systems, with few significant changes during the shoulder seasons apart from the strong positive trends in fall in La Grande Rivière and the Nelson River, and strong (but insignificant) spring discharge increases in the Nelson and Mackenzie Rivers (Fig. 7). In contrast, there is a general trend toward less river discharge during summer with the exception of La Grande Rivière and the Nelson River where diversions from nearby systems enhance flows in all seasons. Chesterfield Inlet exhibits a strong positive trend in summer discharge but again, care must be taken in interpreting this result given the changes in recording methodology in the 1980s (see Sect. 3.1).

## 4.3 Variability arising from flow regulation and climate

Impacts of regulation on discharge variability are examined by considering interdecadal differences in the variability of paired regulated and unregulated rivers (see Sect. 3.2.2). Interdecadal analysis is used because, in some large systems, there has been a stepped introduction of hydroelectric development over the five decades in this study (e.g., Nelson River and La Grande Rivière; see Sect. 2.2). Table 3 presents statistics of interdecadal variability in discharge for eight major regulated rivers and seven of their unregulated counterparts. Regulated rivers in this table are shaded, and their unregulated counterparts are found in the row that follows. Figure 8 shows box and whisker plots of CVs for regulated versus unregulated rivers for each season over the five decades. Table S1 in the online supplement provides the interdecadal variability of all 42 rivers across seasons.

The greatest intradecadal variability is seen from 1974-1983, a period of rapid construction and diversions in most of the regulated rivers (Table 3 and Fig. 8). Flow regulation generally suppresses variability compared to 'matching' unregulated rivers, as has been observed in the Eurasian Ob and Yenisei Rivers (Yang et al., 2004a, b). This effect is greatest post-construction during winter and, to a

5 lesser extent, fall presumably to accommodate higher energy demands (1984-2003; Fig. 8). The Churchill River (Manitoba) is a noteworthy exception to this trend, with increases in interdecadal CV post-diversion (1994-2013 CV = 0.94-1.16) compared to pre-diversion (1964-1973 CV = 0.11). Much of this increase in variability has occurred during fall and winter (1994-2013 CV = 0.94-1.16; Table S1) when mean annual flows are the lowest.

Interestingly, enhanced variability arises in both regulated and unregulated rivers in the most recent decade (2004-2013) for all seasons, but most notably during summer (Fig. 8). The unregulated Nottaway River experiences its highest summer variability (by a factor of two) compared to previous decades (CV = 0.47); and both the regulated Churchill (Manitoba) and Moose Rivers experience their

15 greatest flow variability with their 'matching' unregulated rivers experiencing their second greatest flow variability (Table 3). Increasing overall discharge trends reported in this study are likely influenced by increasing mean summer discharge and variability in the most recent decade (2004-2013), which seems to be largely climate-driven as it occurs in both regulated and unregulated systems. Particularly in the Western Hudson and James Bay region, an increasing number of large summer precipitation and

20 rainfall-runoff events in recent years has, in some cases, yielded annual hydrographs with dual peaks (Ahmari et al., 2016; Blais et al., 2016). Increasing variability may also be influenced by the changing

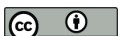



magnitude, timing, frequency and duration of flood events observed in some gauges of our study area (Burn and Whitfield, 2016). Of note is the introduction of regulation on the Rupert River (in 2009), and that enhanced variability from 2004-2013 not seen otherwise in rivers draining to the Eastern Arctic Ocean or Labrador Sea (with the exception of the Alexis River).

## 5 Discussion

### 5.1 Comparison with other studies

Milliman and Farnsworth (2011) provide a comprehensive table of mean annual discharge for most rivers included in this study. Results herein are generally consistent with the mean annual discharge from Milliman and Farnsworth (2011) with variations likely dependent on the selected study period and basin area under consideration. Exceptions arise in some rivers, most notably for Nunavut's Coppermine River where Milliman and Farnsworth (2011) report a mean annual discharge of 11 km$^3$ yr$^{-1}$ and 2.6 km$^3$ yr$^{-1}$ before and after regulation, respectively. Results from this study yield a mean annual discharge of 8.8 km$^3$ yr$^{-1}$ for the Coppermine River – with no known flow regulation over 1964-2013 according to the Water Survey of Canada. Milliman and Farnsworth (2011) also report mean annual discharge rates of 23 km$^3$ yr$^{-1}$ and 15 km$^3$ yr$^{-1}$ before and after regulation, respectively, for northern Ontario's Severn River (despite this system being unregulated), with our results indicating mean annual discharge of 21.9 km$^3$ yr$^{-1}$. The present study also reports underestimates of ~7-13 km$^3$ yr$^{-1}$ for the Albany, Attawapiskat and Winisk Rivers of northern Ontario compared to Milliman and Farnsworth (2011), perhaps owing to different study periods and/or basin areas under consideration. Discrepancies in mean annual discharge for La Grande Rivière, the Eastmain and Rupert Rivers are likely due to recent water management practices to enhance power production at the James Bay

hydroelectric complex. Statistics reported in this study are based on hydrometric data provided directly

by Hydro-Québec (post development of hydropower facilities and infrastructure) that better reflect the

current level of regulation in these systems.

Benke and Cushing (2005) provide mean annual streamflow statistics for 11 rivers examined in this

study (i.e., the Porcupine, Yukon, Chesterfield Inlet, Seal, La Grande, Koksoak, Mackenzie, Moose,

Nelson and Churchill (both in Manitoba and Labrador) Rivers). Our results are generally consistent with

Benke and Cushing (2005) with two notable exceptions. First, mean annual discharge for the Mackenzie

River is ~27 km$^3$ yr$^{-1}$ greater in this study, perhaps due to the exclusion of the Peel River's contribution

to overall Mackenzie River discharge by Benke and Cushing (2005). Second, Benke and Cushing

(2005) report a mean annual discharge of ~200 km$^3$ yr$^{-1}$ for the Yukon River, although this considers

both the Canadian and American contributing area (total of 839,200 km$^2$), rather than just the upstream

part in Canada examined here (gauged area = 288,000 km$^2$). Discrepancies also arise for the Koksoak

River and La Grande Rivière as mean annual discharge values reported by Benke and Cushing (2005)

reflect conditions prior to the diversion of the upper Caniapiscau River to the La Grande Rivière system.

While Déry and Wood (2005) first reported a 10% decline in river discharge across northern Canada

from 1964 to 2003, this effort finds a remarkable reversal to that trend in expanding the study period by

only a decade. In the second half of the study period, river discharge in northern Canada increased by

18.1% (relative to its overall mean annual discharge). This is in stark contrast to the first half of the

study period during which river discharge declined significantly. Rood et al. (2016) also documented a

statistically-significant ($p < 0.05$) ~1.5% (decade)$^{-1}$ increase in Mackenzie River discharge over 1939-2013, consistent with the pattern observed in this study; however, our abbreviated study period reveals an insignificant trend in this system (Table 1). These findings suggest that rivers in northern Canada are now responding similarly to rising air temperatures as those in Eurasia (Peterson et al., 2002), in accord

with climate change projections (Milly et al., 2005). In fact, the rate of increase of 8.4 km$^3$ yr$^{-1}$ yr$^{-1}$ for river discharge across northern Canada from 1989 to 2013 exceeds the overall trend of 6.3 km$^3$ yr$^{-1}$ yr$^{-1}$ for 16 Eurasian rivers draining to the Arctic Ocean from 1964 to 2000 (McClelland et al., 2006). The period of record remains relatively short and care must be taken in interpreting these findings; nonetheless attempts should be made to reconcile observed trends in northern Canada and Eurasia using

identical study periods and methodologies. Decadal climate variability associated with the Arctic, Pacific Decadal, and Atlantic Multidecadal Oscillations, among other large-scale modes of climate variability, are known to affect river discharge in northern Canada (Déry and Wood, 2004; Kingston et al., 2006; Assani et al., 2010; Rood et al., 2016). We stress that it remains important to continue to monitor river discharge in northern Canada (and expand where possible to attain WMO standards) as

this region is expected to continue warming rapidly in the twenty-first century (Coulibaly et al., 2013; Gough and Wolfe, 2001; Gagnon and Gough, 2005).

## 5.2 Anthropogenic influences

While rising air temperatures and changing precipitation patterns are key factors in altering northern

Canadian river discharge, another control remains anthropogenic activities such as water retention, regulation, and diversion. The development of large hydroelectric complexes in northern Québec, Ontario, Manitoba and across the Canadian Prairies into BC has significantly altered the seasonality of

flows in northern Canada, most notably to Western and Eastern Hudson and James Bay. River

diversions and flow regulation typically do not influence overall flow volumes to the coastal ocean

(McClelland et al., 2006); however, pronounced changes in seasonality accompany regulation,

especially in systems with large storage capacity such as the Mackenzie (Peace) River and La Grande

Rivière. Furthermore, short term (i.e. 1-5 years) declines in river discharge to the coastal ocean can arise

from the filling of large reservoirs for hydropower production. Across northern Canada, >300 km$^3$ of

water storage capacity has been developed since 1964 (Lee et al., 2012). This may lead to both a short

term decline in observed flows to the coastal ocean while reservoirs are filled, and 'aging' of water in

storage affecting its properties such as biochemistry and temperature while enhancing evaporative

losses (Vörösmarty and Sahagian, 2000). Perhaps lesser known are the impacts of land cover and land

use change on river discharge in northern Canada. Deforestation through wood harvesting depresses

water demand by vegetation while increasing soil moisture and runoff generation (Boon, 2012). In

contrast, the intensification of agricultural activities, particularly in the Canadian Prairies, increases

water demand for irrigation. Anthropogenic activities play a major role in changing pan-Arctic

hydrology that requires special attention when assessing observed changes.

## 5.3 Physical impacts to the marine environment

Changes in seasonal river conditions induced by climate change and river regulation affect the physical

regime of coastal estuaries by modifying salinity levels and the input of nutrients and sediments

(Gillanders et al., 2011). For instance, the Eastmain River estuary experienced an increase in salinity

after diversion into La Grande Rivière system (Messier et al., 1986; Drinkwater and Frank, 1994);

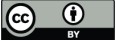

concurrently lowering salinity in La Grande Rivière's estuary during winter (Whittaker, 2006). Increasing river discharge in northern Canada strengthens ocean stratification, thereby suppressing deep-water formation in the Labrador Sea (Myers, 2005). This, in turn, weakens the thermohaline circulation that is responsible for the transport of heat and nutrients in the North Atlantic Ocean (Ogi et

al., 2001; Rennermalm et al., 2007). Increasing winter discharge in northern Canada also delivers sensible heat to the coastal ocean, promoting the ablation of sea ice in estuaries and delta regions (Kuzyk et al., 2008; Searcy et al., 1996). This emphasizes the need for basin-scale numerical modelling of the coupling between freshwater fluxes with the marine environment (e.g., Saucier et al., 2004).

**6 Conclusion**

This study provides an update on the recent variability and trends in river discharge across northern Canada. In contrast to previous studies (e.g., Déry and Wood, 2005; McClelland et al., 2006), we report a strong increasing trend in river discharge in northern Canada since the 1990s. Between 1989 and 2013, total annual river discharge in northern Canada increased by 208.8 km$^3$ (25 year)$^{-1}$, equivalent to

15 an 18.1% rise relative to mean annual discharge over the study period. This aligns with recent trends in Eurasia associated with increased moisture transport to high northern latitudes (Zhang et al., 2013; Rawlins et al., 2009). The recent tendency towards a negative phase of the Arctic Oscillation, perhaps associated with declining Arctic sea ice, is likely contributing to increasing river discharge in northern Canada (Déry and Wood, 2004; Screen et al., 2013). The positive phase of the Arctic Oscillation

advects cold, dry air over northeastern Canada, reducing snowfall amounts and river discharge (Déry and Wood, 2004; Déry et al., 2005b). Thus warming air temperatures and reductions in Arctic sea ice

extent result in more abundant precipitation across northern Canada that yield higher discharge rates to the Arctic Ocean and adjacent northern seas. An avenue for future work therefore will be spectral and/or wavelet analysis of river discharge records in northern Canada for comparison with climate variability associated with large-scale teleconnections; such as the Arctic, Pacific Decadal and Atlantic

Multidecadal Oscillations. Isolating the impacts of large-scale climate variability on river discharge through such methods would facilitate more robust detection of linear trends in the hydrological records associated with climate warming, among other factors.

Flow regulation is shown to suppress natural discharge variability, particularly during winter. This

effect is distinguishable in flow records following the period of intensive construction of hydroelectric facilities in northern Canada (mid-1970s to early-1980s). Of note is the augmented variability in both regulated and unregulated rivers during the most recent decade of study (2004-2013), which may be climate-driven.

Since the approach used in this study relies on the existence of observed hydrometric data, there are temporal and spatial gaps in our dataset and analyses, most notably in the high Arctic and CAA (Fig. 1). Temporal gaps are in-filled using a two-step approach that includes use of daily climatological values of streamflow when data from upstream gauges remain unavailable, with the caveat of possibly reducing the magnitude and significance of monotonic trends. A future effort will therefore refine this strategy by

considering linear interpolation for short (≤1 week) temporal gaps and cross correlations or Hirsch's (1982) maintenance of variance method (MOVE) based on proximal rivers for longer (>1 week) periods

of missing data (Hernández-Henríquez et al., 2010), among other methods. An additional approach to in-fill both temporal and spatial gaps consists of hydrological modelling combined with meteorological forcing from observational, reanalysis or modelling datasets. Use of a hydrological model forced by output from global climate models under various scenarios also allows projections of future discharge

across northern Canada. This work is currently being undertaken by the authors with the Arctic-HYPE hydrological model (Andersson et al., 2015) for the Hudson and James Bay drainage basin.

*Acknowledgements*. Thanks to the Water Survey of Canada, Manitoba Hydro, Hydro-Québec, and the Direction d'Expertise Hydrique du Québec for access to hydrometric data and the Natural Sciences and

10 Engineering Research Council of Canada, Manitoba Hydro, and partners through funding of the BaySys project. Thanks to Marco Hernández-Henríquez (UNBC) for assistance in figure preparation for the poster that preceded this manuscript, Shane Wruth (Manitoba Hydro) and Catherine Guay (Hydro-Québec) for compiling data for the Nelson River and La Grande Rivière, respectively, Kristina Koenig and colleagues at Manitoba Hydro for logistical support and review of the paper, and to Eric Wood

(Princeton University) for motivating this effort and ongoing support of this research.

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





**Table 1.** List of 42 rivers (from west to east) and their tributaries (italicized) that discharge into six drainage basins of northern Canada with geographical coordinates of the recording gauge nearest to the mouth, contributing area that is gauged, the annual mean, standard deviation (SD), coefficient of variation (CV), and trend, 1964-2013.

| Region | River | Lat (°N) | Lon (°W) | Gauged Area (km²) | Mean (km³ yr⁻¹) | SD (km³ yr⁻¹) | CV | Trend (km³ 50 yr⁻¹) |
|---|---|---|---|---|---|---|---|---|
| **Bering Sea** | Yukon | 64.79 | 141.20 | 288,000 | 77.28 | 9.48 | 0.12 | 3.23 |
| | Porcupine | 67.42 | 140.89 | 58,900 | 10.57 | 2.49 | 0.24 | -1.26 |
| **Western Arctic Ocean** | Firth | 69.33 | 139.57 | 5,700 | 1.21 | 0.23 | 0.19 | -0.02 |
| | Mackenzie *Mackenzie* *Peel* | 67.46 67.24 | 133.75 134.89 | 1,749,700 1,679,100 70,600 | 311.38 | 32.18 | 0.10 | 20.87 |
| | Anderson | 68.63 | 128.42 | 57,800 | 4.72 | 1.39 | 0.30 | 0.06 |
| | Coppermine | 67.23 | 115.89 | 46,200 | 8.77 | 1.64 | 0.19 | -0.73 |
| | Tree | 67.64 | 111.90 | 5,810 | 1.11 | 0.24 | 0.22 | -0.05 |
| | Burnside | 66.73 | 108.81 | 16,800 | 4.20 | 0.98 | 0.23 | 0.06 |
| | Ellice | 67.71 | 104.14 | 16,900 | 2.82 | 0.64 | 0.23 | 0.08 |
| | Back | 66.09 | 96.51 | 93,900 | 15.52 | 3.17 | 0.20 | -0.40 |
| | Freshwater Creek | 69.13 | 104.99 | 1,490 | 0.14 | 0.04 | 0.26 | 0.00 |
| **Western Hudson and James Bay** | Chesterfield Inlet *Thelon* *Kazan* | 64.77 63.65 | 97.05 95.08 | 224,000 154,000 70,000 | 41.28 | 6.93 | 0.17 | 4.86 |
| | Thlewiaza | 60.78 | 98.77 | 27,000 | 6.82 | 0.81 | 0.12 | 0.00 |
| | Seal | 58.89 | 96.27 | 48,200 | 11.49 | 2.46 | 0.21 | 1.15 |
| | Churchill *Churchill* *Deer* | 58.12 58.01 | 94.62 94.19 | 290,880 289,000 1,880 | 18.90 | 13.25 | 0.70 | -9.31* |
| | Nelson *Angling* *Limestone* *Nelson* *Weir* | 56.67 56.51 56.37 57.20 | 93.64 94.21 94.63 93.45 | 1,125,520 1,560 3,270 1,100,000 2,190 | 102.70 | 22.63 | 0.22 | 18.70 |
| | Hayes | 56.43 | 92.79 | 103,000 | 19.71 | 4.96 | 0.25 | -2.32 |
| | Severn | 55.37 | 88.32 | 94,300 | 21.90 | 5.59 | 0.26 | 0.02 |
| | Winisk *Shamattawa* | 54.28 | 85.65 | 54,710 4,710 | 15.24 | 4.69 | 0.31 | -2.32 |





| | | | | | | | |
|---|---|---|---|---|---|---|---|
| | *Winisk* | 54.52 | 87.23 | 50,000 | | | | |
| | Ekwan | 53.80 | 84.92 | 16,900 | 2.76 | 0.69 | 0.25 | 0.00 |
| | Attawapiskat | 53.09 | 85.01 | 36,000 | 11.43 | 3.33 | 0.29 | -1.54 |
| | Albany | 51.33 | 83.84 | 118,000 | 31.77 | 8.06 | 0.25 | 2.14 |
| | Moose | | | 98,530 | | | | |
| | *Abitibi* | 50.60 | 81.41 | 27,500 | | | | |
| | *Kwataboa-hegan* | 51.16 | 80.86 | 4,250 | 39.01 | 7.28 | 0.19 | -6.58* |
| | *Moose* | 50.81 | 81.29 | 60,100 | | | | |
| | *North French* | 51.07 | 80.76 | 6,680 | | | | |
| **Eastern Hudson and James Bay** | Harricana | | | 21,200 | | | | |
| | *Harricana* | 49.95 | 78.72 | 10,000 | 7.75 | 1.00 | 0.13 | 0.02 |
| | *Turgeon* | 49.98 | 79.09 | 11,200 | | | | |
| | Nottaway | 50.13 | 77.42 | 57,500 | 32.27 | 5.32 | 0.16 | -2.79 |
| | Broadback | 51.18 | 77.43 | 17,100 | 10.03 | 1.53 | 0.15 | 0.66 |
| | Rupert | 51.44 | 76.86 | 40,900 | 25.32 | 4.93 | 0.19 | -2.99 |
| | Pontax | 51.53 | 78.09 | 6,090 | 3.12 | 0.37 | 0.12 | 0.00 |
| | Eastmain | 52.24 | 78.07 | 44,300 | 12.11 | 12.73 | 1.05 | -0.63 |
| | La Grande | 53.72 | 78.57 | 96,600 | 84.22 | 24.38 | 0.29 | 14.27* |
| | Grande Rivière de la Baleine | 55.29 | 77.59 | 43,200 | 19.61 | 2.60 | 0.13 | -3.78* |
| | Nastapoca | 56.86 | 76.21 | 12,500 | 7.94 | 0.91 | 0.11 | 0.00 |
| **Eastern Arctic Ocean** | Aux Feuilles | 58.64 | 70.42 | 41,700 | 17.62 | 2.14 | 0.12 | -0.40 |
| | Koksoak | | | 127,200 | | | | |
| | *Caniapiscau* | 57.42 | 69.25 | 84,500 | 55.57 | 14.76 | 0.27 | -7.01 |
| | *Aux Mélèzes* | 58.64 | 70.42 | 42,700 | | | | |
| | À la Baleine | 57.88 | 67.58 | 29,800 | 16.02 | 1.96 | 0.12 | -1.74* |
| | George | 58.15 | 65.84 | 35,200 | 23.73 | 3.07 | 0.13 | -0.33 |
| | Sylvia Grinnell | 63.77 | 68.58 | 2,980 | 1.07 | 0.21 | 0.20 | 0.00 |
| **Labra-dor Sea** | Naskaupi | 54.13 | 61.43 | 4,480 | 6.00 | 1.02 | 0.17 | -1.15* |
| | Churchill | 53.25 | 60.79 | 92,500 | 56.29 | 7.20 | 0.13 | -2.58 |
| | Eagle | 53.53 | 57.49 | 10,900 | 8.02 | 1.29 | 0.16 | -0.47 |
| | Alexis | 52.65 | 56.87 | 2,310 | 1.66 | 0.22 | 0.13 | 0.00 |
| | Ugjoktok | 55.23 | 61.30 | 7,570 | 5.06 | 0.74 | 0.15 | -0.07 |

*Statistically-significant trends ($p<0.05$).




**Table 2.** Statistics of gauged area, mean, standard deviation (SD), coefficient of variation (CV) of annual discharge, and 50-year discharge trend for six drainage basins in northern Canada, 1964-2013.

| Region | Area (km$^2$) | Regulated Area (%) | Mean (km$^3$ yr$^{-1}$) | SD (km$^3$ yr$^{-1}$) | CV | Trend (km$^3$ 50 yr$^{-1}$) |
|---|---|---|---|---|---|---|
| Bering Sea | 346,900 | 0.0 | 87.8 | 10.0 | 0.11 | 2.1 |
| Western Arctic Ocean | 1,998,800 | 14.6* | 349.9 | 32.8 | 0.09 | 26.2 |
| Western Hudson Bay | 2,220,400 | 63.0** | 323.0 | 41.2 | 0.13 | 9.3 |
| Eastern Hudson Bay | 333,070 | 55.3** | 202.4 | 20.5 | 0.10 | 24.9[#] |
| Eastern Arctic Ocean | 233,900 | 36.1 | 114.0 | 18.7 | 0.16 | -38.1[#] |
| Labrador Sea | 122,270 | 75.7 | 77.0 | 8.7 | 0.11 | -0.05 |
| All Regions | 5,255,340 | 40.9** | 1154.1 | 76.1 | 0.07 | 10.5 |

*Covers only area of the Peace River Basin, the remainder of the Mackenzie River Basin is assumed to be unregulated.

**As of 2013.

[#]Statistically-significant trends ($p<0.05$).





**Table 3.** Paired comparisons of decadal statistics for the mean annual discharge, SD and CV of regulated and matching unregulated rivers.

| Rivers* | 1964-1973 | | | 1974-1983 | | | 1984-1993 | | | 1994-2003 | | | 2004-2013 | | |
|---|---|---|---|---|---|---|---|---|---|---|---|---|---|---|---|
| | Mean (km³ yr⁻¹) | SD (km³ yr⁻¹) | CV | Mean (km³ yr⁻¹) | SD (km³ yr⁻¹) | CV | Mean (km³ yr⁻¹) | SD (km³ yr⁻¹) | CV | Mean (km³ yr⁻¹) | SD (km³ yr⁻¹) | CV | Mean (km³ yr⁻¹) | SD (km³ yr⁻¹) | CV |
| Churchill (Manitoba) | 37.00 | 4.20 | 0.11 | 23.70 | 13.49 | 0.57 | 8.43 | 2.93 | 0.35 | 9.59 | 4.40 | 0.46 | 15.76 | 10.50 | 0.67 |
| Nelson | 90.42 | 15.23 | 0.17 | 94.90 | 14.26 | 0.15 | 91.84 | 15.92 | 0.17 | 105.55 | 19.33 | 0.18 | 130.77 | 21.79 | 0.17 |
| Seal | 10.91 | 2.58 | 0.24 | 11.76 | 3.29 | 0.28 | 11.35 | 1.48 | 0.13 | 11.08 | 2.17 | 0.20 | 12.35 | 2.67 | 0.22 |
| Hayes | 21.55 | 2.99 | 0.14 | 20.43 | 3.85 | 0.19 | 16.63 | 5.42 | 0.33 | 18.63 | 4.87 | 0.26 | 21.31 | 6.16 | 0.29 |
| La Grande | 56.81 | 6.93 | 0.12 | 58.77 | 9.62 | 0.16 | 96.11 | 11.37 | 0.12 | 99.50 | 11.36 | 0.11 | 109.89 | 11.51 | 0.10 |
| Grande Rivière de la Baleine | 21.86 | 2.11 | 0.10 | 20.04 | 2.25 | 0.11 | 19.38 | 2.45 | 0.13 | 17.80 | 2.85 | 0.16 | 18.98 | 1.80 | 0.09 |
| Eastmain | 30.01 | 3.72 | 0.12 | 20.62 | 13.12 | 0.64 | 3.17 | 0.34 | 0.11 | 3.11 | 0.39 | 0.13 | 3.62 | 0.06 | 0.02 |
| Rupert | 27.54 | 3.51 | 0.13 | 27.65 | 2.98 | 0.11 | 25.18 | 1.83 | 0.07 | 26.17 | 2.28 | 0.09 | 20.04 | 7.70 | 0.38 |
| Nottaway | 33.11 | 4.71 | 0.14 | 33.49 | 5.31 | 0.16 | 29.93 | 6.84 | 0.23 | 32.97 | 4.54 | 0.14 | 31.84 | 5.20 | 0.16 |
| Koksoak | 74.90 | 8.27 | 0.11 | 66.21 | 12.96 | 0.20 | 45.58 | 7.85 | 0.17 | 44.51 | 2.51 | 0.06 | 46.65 | 2.38 | 0.05 |
| A la Baleine | 17.07 | 2.31 | 0.14 | 16.95 | 1.49 | 0.09 | 14.60 | 2.52 | 0.17 | 15.57 | 1.06 | 0.07 | 15.91 | 1.06 | 0.07 |
| Moose | 42.76 | 6.05 | 0.14 | 39.45 | 7.63 | 0.19 | 38.77 | 6.28 | 0.16 | 35.91 | 4.31 | 0.12 | 38.16 | 10.39 | 0.27 |
| Albany | 33.26 | 6.90 | 0.21 | 28.40 | 7.08 | 0.25 | 30.04 | 10.88 | 0.36 | 32.14 | 1.40 | 0.04 | 34.98 | 10.26 | 0.29 |
| Churchill (Labrador) | 51.56 | 8.39 | 0.16 | 64.63 | 6.79 | 0.10 | 53.44 | 5.39 | 0.10 | 56.02 | 4.14 | 0.07 | 55.78 | 2.84 | 0.05 |
| Eagle | 8.02 | 1.14 | 0.14 | 8.77 | 1.66 | 0.19 | 7.48 | 1.39 | 0.19 | 8.05 | 1.08 | 0.13 | 7.79 | 0.95 | 0.12 |

* Shading denotes regulated rivers




# Figures

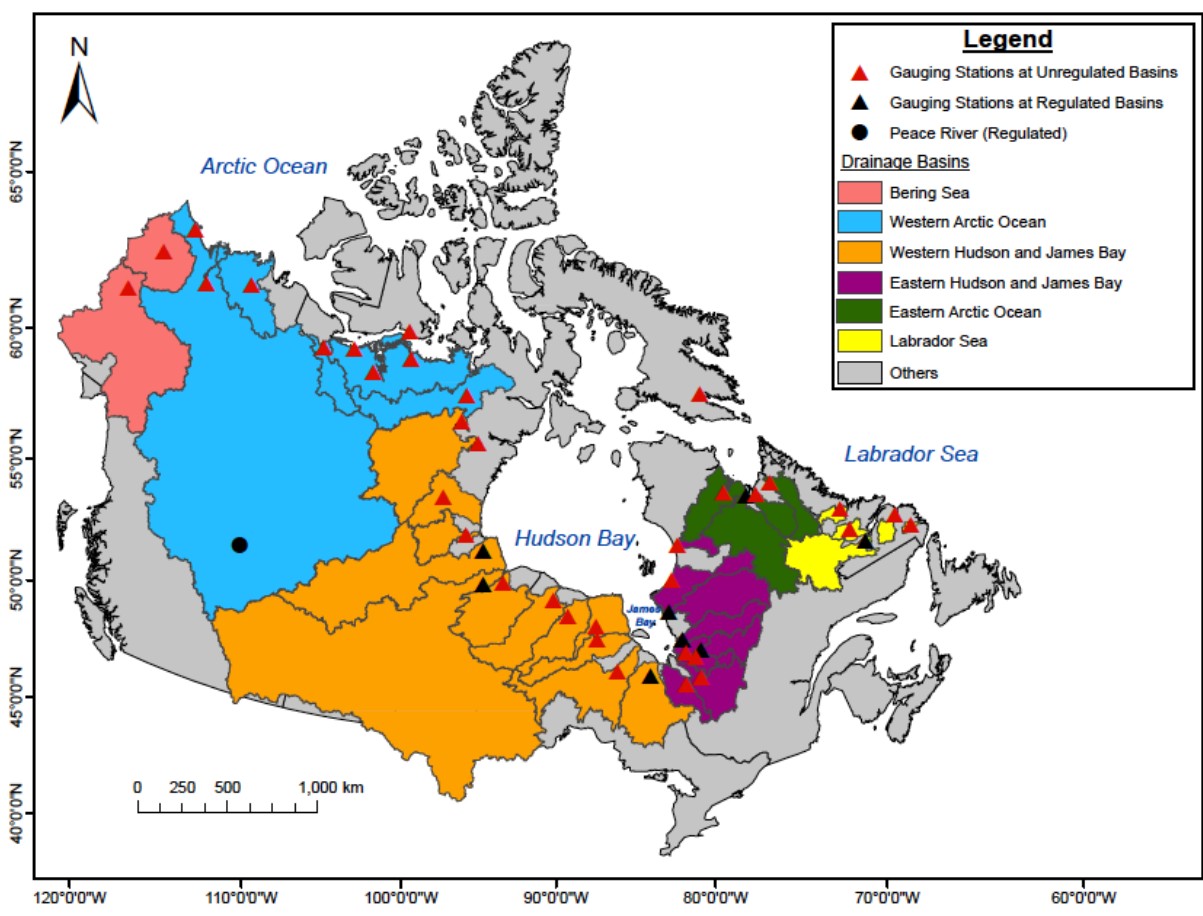

**Figure 1.** Map of the six major basins draining northern Canada and parts of the northern United States
as well as the spatial distribution of hydrometric gauges used in this study.





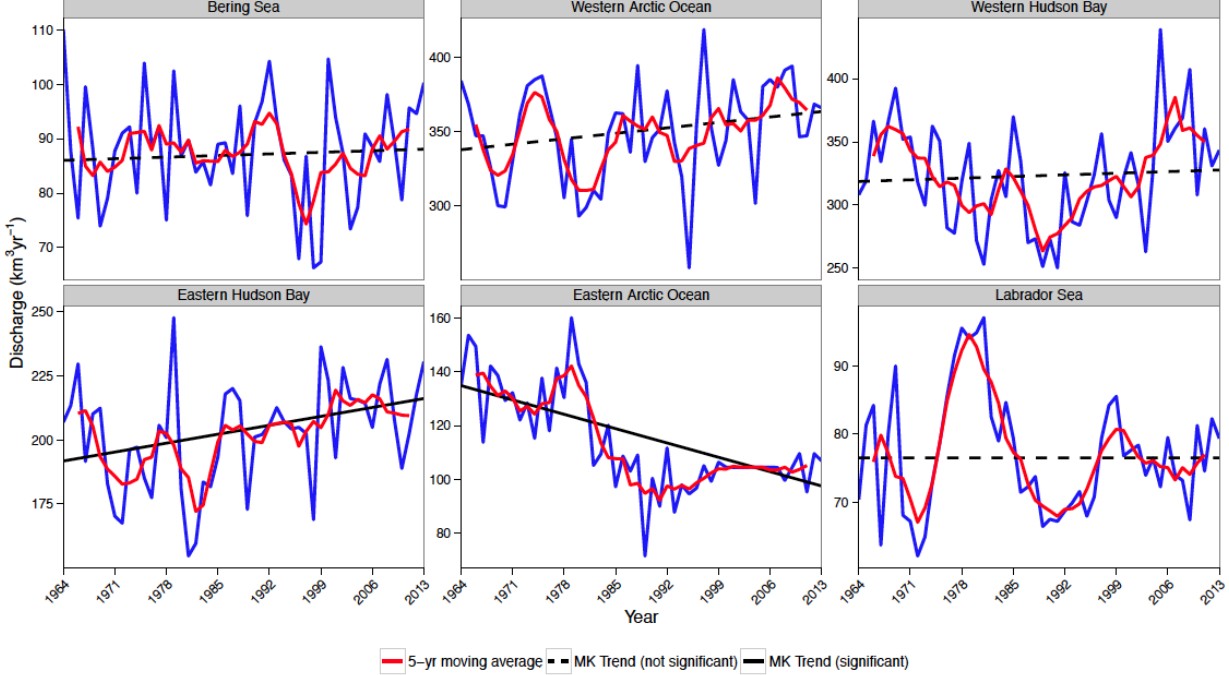

**Figure 2.** Time series of total annual discharge for six major drainage basins of northern Canada, 1964-2013.



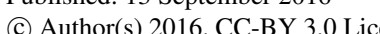


**Figure 3.** Time series of total annual discharge for 42 rivers draining northern Canada, 1964-2013.




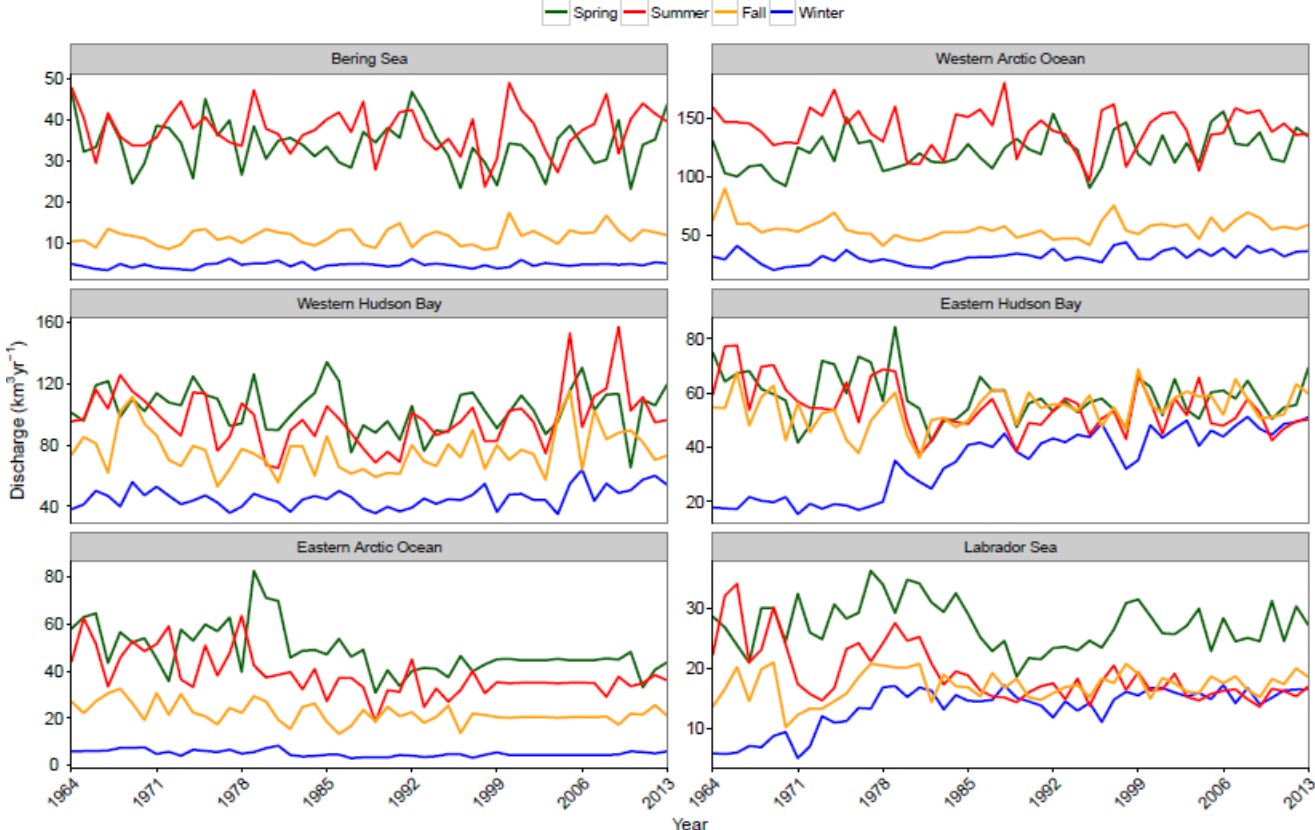

**Figure 4.** Time series of total seasonal discharge for six major drainage basins of northern Canada, 1964-2013.





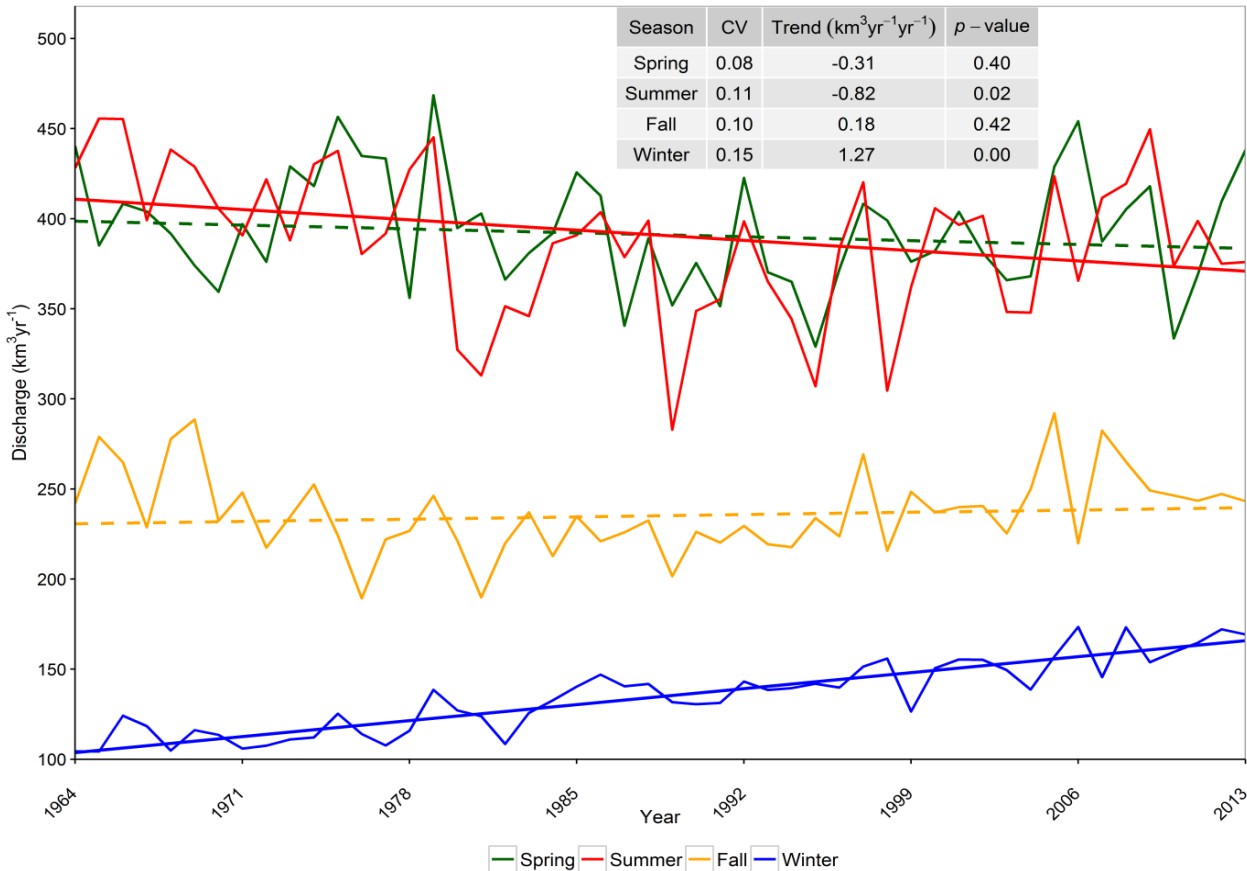

**Figure 5.** Time series of total seasonal discharge for 42 rivers draining northern Canada, 1964-2013.





**Figure 6.** Spatial trend analysis for the annual discharge of 42 rivers of northern Canada, 1964-2013.







**Figure 7.** Spatial trend analysis for the seasonal discharge of 42 rivers of northern Canada, 1964-2013.



**Figure 8.** Box and whisker plots of the coefficient of variation of seasonal river discharge for eight regulated and seven 'matching' unregulated rivers for five decades over 1964-2013. Boxes indicate the 25th and 75th interquantile ranges with the central horizontal lines denoting the median values, while the whiskers represent the 5th and 95th percentiles in the data.