# Peer review of "Recent trends and variability in river discharge across northern Canada"

_Hydrology and Earth System Sciences, 2016_

## Referee Comment (RC2)

**General Comments:**

This manuscript, titled "Recent trends and variability in river discharge across northern Canada", summarizes interdecadal trends and interannual variability in river discharge for 42 principal rivers draining northern Canada over a 50-year period. The research focused on the science question "whether or not river discharge in northern Canada shows a continued decrease in the twenty-first century as first reported in the leading author's previous publication". Also, the effects of flow regulation and climate variability are both considered in the manuscript. The leading author has much experience on the similar topic. Also, the manuscript is clearly written. Therefore, the reviewer would recommend this manuscript for publication.

Moreover, the reviewer would suggest the following specific comments to improve the clarity of the paper.

**Specific Comments:**

1)  In northern Canada, there are substantial gaps in the time series of river discharge. The authors used a two-step process similar to Déry et al. (2005a). However, the streamflow timing change was not considerd in this gap filling strategy. As indicated in the manuscript, the gap-filling process can influence the magnitude of MKT trends. Therefore, the uncertainty resulting from streamflow timing change should be included in Section 5 (Discussion).

2)  For some river basins affected by seasonally frozen soils and permafrost in northern Canada, there are significantly increased groundwater to river discharge due to the thawing of seasonally frozen soils and permafrost. For their contribution in trends and variability of river discharge, the authors mentioned in the text but without further discussion. Also, the related references are from 1986 (Woo) to 2007 (Walwoord and Striegl), which need to be updated and be consistent with your analysis period.

3)  Although the effects of flow regulation and climate variability were both discussed in the manuscript, their effects were not split clearly in the analyses. It would be helpful to use the modelling tool and separate their effects as part of your future work. Please discuss this in Section 5.

---

## Author Comment (AC1) · 16 Sep 2016

Please note that a supplement was added to this submission two days after the initial publication of the discussion paper. This brief delay in the submission of the supplementary materials occurred due to issues incurred during the initial upload of the documents. The supplementary document contains only one extended table (Table S1) with individual statistics of the mean, standard deviation and coefficient of variation in annual river discharge over five decades for the 42 rivers under study. Highly fragmented and regulated systems are highlighted in gray in the table. Comments on this supplementary table and the discussion paper itself are welcomed by the authors.

---

## Referee Comment (RC1) · Anonymous Referee #1 · 11 Oct 2016

This manuscript, based on the lead author's previous work and experience, re-evaluates river discharge trends across northern Canada. It examines long-term (50 yrs) and large-scale changes/variability in discharge for 42 rivers draining northern Canada. The research question of this paper is important, i.e. whether or not river discharge in northern Canada has a continued decrease in the twenty-first century as first reported by Déry and Wood (2005). This work also discusses the effects of flow regulation and climate variability over the large regions. The data, methods, and results of this analysis are useful for cold region hydrology/climate and arctic research.

This paper is clearly written and easy to follow and read. I have some comments and suggestions below and would recommend a minor revision to further improve the quality of this manuscript.

Dam effect and regulation: Flow regulation by dams and reservoirs exist in most of the large northern basins in Canada. Many studies clearly show that dam regulation significantly affects the timing and magnitude of discharge at season to annual time scales. This paper separated basins with and without dams and compared their flow changes and variations. To better understand dam effects in this analysis, it is necessary to collect and present the dam info for each basin. I would suggest providing dam info in Table 1, including number of dams in the basins and total capacity of all reservoirs relative to the basin mean flow. This will be a useful measure to reflect the degree of regulation for a specific watershed. With this info for all the regulated basins, it is then possible and useful to integrate it into the regional scale and discussion of the flow CV and change.

This paper discussed the regional flow regimes without the use of any hydrographs. The authors do have the daily and monthly flow records for each basin. It is very useful to show the mean hydrograph for the regions in discussion. With the hydrographs, monthly flow data will be used and presented to show the seasonality of flow and also the difference of mean flows between the regulated and unregulated basins. I would also suggest, for the regulated basins, to work out and compare the mean flows for the pre- and post-dam periods, so as to demonstrate the magnitude of dam effects on flow conditions.

Citation of papers: Rivers are very different from one to the other. Many good papers have been published in the last 10 years on northern river hydrology; some of them are extremely useful for this work over northern Canada. The authors should read and cite more papers (a short list below), particularly those on basin-scale analyses of hydrological regimes and changes induced by climate variation and human impacts.

Woo, M.K., Thorne, R., 2003: Streamflow in the Mackenzie Basin, Canada. Arctic 56, 328-340.

Peters, D.L., Prowse, T.D., 2001: Regulation effects on the lower Peace River, Canada.
Hydrol. Process. 15, 3181e3194.

Yang, D., X. Shi, P. Marsh, 2014: Variability and extreme of Mackenzie River daily discharge during, 1973-2011, Quaternary International, doi/10.1016 /j.quaint.2014.09.023

Wang S. J. Huang, D. Yang, G. Pavlic, and J. Li, 2014: An assessment of long-term water budget closures for large drainage basins in Canada, Hydrological Processes, DOI: 10.1002/hyp.10343.

Conclusion: The paper states. . ." Of note is the augmented variability in both regulated and unregulated rivers during the most recent decade of study (2004-2013), which may be climate-driven. It is clear from many other studies that the regulated basins are not reliable to reflect climate change impact to basin hydrology change. The argument in this paper about climate-driven similar changes in flow variation in regulated and unregulated basins might be true; however, the fact remains that with the strong dam regulation, it is difficult or impossible to detect climate signal over a basin. Thus, it is always better to rely on flow data and info from the unregulated rivers.

Finally, at the end of the Conclusion Section, the paper states "Use of a hydrological model forced by output from global climate models under various scenarios also allows projections of future discharge across northern Canada. This work is currently being undertaken by the authors with the Arctic-HYPE hydrological model (Andersson et al., 2015) for the Hudson and James Bay drainage basin." This work is interesting, but it is unclear if the ongoing modeling work has considered human activities, particularly dams in the basins.

---

## Author Comment (AC2) · 2 Nov 2016

We sincerely thank Anonymous Referees #1 and #2 for their constructive comments on our manuscript (Reference # HESS-2016-461). We fully recognize and appreciate the reviewers' efforts in providing these informative reports on our research focused on river discharge in northern Canada. Indeed, their insights are undoubtedly leading to an improved paper through this online discussion and ensuing revision process. We are thus taking into full consideration all of the comments from the referees and are preparing detailed responses to these as well as information on how the paper is being revised according to the two anonymous referees' suggestions. A complete and detailed response document will be submitted once a decision has been made on our discussion paper. In the meantime, we provide here a general overview of our responses to the comments submitted by each referee:

Anonymous Referee #1:

First, we are updating Table 1 in our paper to include information on the number of reservoirs and their total storage capacity in each of the river basins under study. This is particularly important in interpreting seasonal trends in river discharge to the coastal ocean in systems such as the Nelson and La Grande Rivière that are highly fragmented. Second, the construction of the river discharge time series is based on daily hydrometric data. As such it is straightforward to include climatological hydrographs for the six regions of interest. To that end, the new section 4.4 will provide the mean annual cycles of daily river discharge for the six regional basins under study as well as comparisons of hydrographs between regulated and unregulated systems. Third, we acknowledge that some important references on the variability and trends in streamflow across northern Canada are absent in our paper. Those suggested by the referee and a few other recent publications are now being added to our revised manuscript (see below). Fourth, we agree with the referee that climate change signals are more robust and easier to detect in unregulated rivers. A statement to that effect is now being included in the Conclusion. Finally, we are clarifying the text in the Conclusion describing ongoing modeling work the Arctic-HYPE model, which is indeed considering human activities (and climate change) on flows reaching polar seas.

Anonymous Referee #2:

The first comment from Referee #2 relates to the strategy used to in-fill gaps in the time series of river discharge in northern Canada, namely streamflow timing changes and their impacts on filling missing daily discharge data. Inclusion of this possible factor on the trend analysis is now being added to Section 3.2.2 in the Methods where caveats of our gap-filling strategy on our analyses are discussed. While the referee recommends inclusion of this point in Section 5 (Discussion), there is no discussion on some of the assumptions or potential shortcomings of the gap-filling strategy there and impacts on the trend analyses. As such, this comment is being addressed in the second paragraph of Section 3.2.2. Permafrost degradation is a possible factor influencing observed streamflow in northern Canada (St. Jacques and Sauchyn 2009). Further text is therefore being added in the paper given the possible impacts of permafrost thaw in northern Canada's hydrology. Identification of the effects of flow regulation and climate variability and change on streamflow in northern Canada remains a goal of ongoing research. Section 5 now therefore includes a brief summary of an ongoing modelling exercise aimed at attributing the impacts of both climate change and anthropogenic activities on streamflow input to Hudson Bay.

New References:

St. Jacques, J.-M. and Sauchyn, D. J.: Increasing winter baseflow and mean annual streamflow from possible permafrost thawing in the Northwest Territories, Canada, Geophys. Res. Lett., 36, L01401, doi: 10.1029/2008GL035822, 2009.

Tananaev, N. I., Makarieva, O. M., and Lebedeva, L. S.: Trends in annual and extreme flows in the Lena River basin, Northern Eurasia, Geophys. Res. Lett., 43, doi: 10.1002/2016GL070796, 2016.

van Vliet, M. T. H., Franssen, W. H. P., Yearsley, J. R., Ludwig, F., Haddeland, I., Lettenmaier, D. P., and Kabat, P.: Global river discharge and water temperature under climate change, Glob. Planet. Change, 23, 450-464, 2013.

---

## Author Response (AR1)

**RESPONSE TO THE REFEREES' COMMENTS**

Dear Dr. Lettenmaier:

We sincerely thank both referees for their thorough reviews and most constructive comments on our manuscript (Reference # HESS-2016-461). We fully recognize and appreciate the reviewers' efforts in providing these informative reports on our research and their insights have led to an improved interpretation of our results. We have therefore taken into full consideration all of these comments and have prepared responses to these as well as information on how the paper was revised following the referees' suggestions. Our responses and edits to the paper are provided below **in bold** following the individual comments requiring action from both reviewers.

Please do not hesitate to contact us if any of this information is not clear.

With kind regards,
Stephen Déry

Referee #1:

This manuscript, based on the lead author's previous work and experience, reevaluates river discharge trends across northern Canada. It examines long-term (50 yrs) and large-scale changes/variability in discharge for 42 rivers draining northern Canada. The research question of this paper is important, i.e. whether or not river discharge in northern Canada has a continued decrease in the twenty-first century as first reported by Déry and Wood (2005). This work also discusses the effects of flow regulation and climate variability over the large regions. The data, methods, and results of this analysis are useful for cold region hydrology/climate and arctic research.

This paper is clearly written and easy to follow and read. I have some comments and suggestions below and would recommend a minor revision to further improve the quality of this manuscript.

**Thank you kindly for this summary of our paper. Indeed, this effort builds on the lead author's previous research on river discharge in northern Canada by extending the study period by at least a decade and to add a focus on streamflow regulation impacts to the observed variability and trends in flows.**

Dam effect and regulation: Flow regulation by dams and reservoirs exist in most of the large northern basins in Canada. Many studies clearly show that dam regulation significantly affects the timing and magnitude of discharge at season to annual time scales. This paper separated basins with and without dams and compared their flow changes and variations. To better

understand dam effects in this analysis, it is necessary to collect and present the dam info for each basin. I would suggest providing dam info in Table 1, including number of dams in the basins and total capacity of all reservoirs relative to the basin mean flow. This will be a useful measure to reflect the degree of regulation for a specific watershed. With this info for all the regulated basins, it is then possible and useful to integrate it into the regional scale and discussion of the flow CV and change.

**Many of the rivers in northern Canada are influenced by flow regulation through infrastructure such as hydropower dams and reservoirs. In response to this comment from Referee #1, Table 1 now includes the number of reservoirs and their volumetric capacity (sourced from the Global Reservoir and Dams (GRanD) database by Lehner et al. (2008)) for each of the river basins under study.**

**New Reference:**

**Lehner, B., Reidy Liermann, C., Revenga, C., Vörösmarty, C., Fekete, B., Crouzet, P., Döll, P., Endejan, M., Frenken, K., Magome, J., Nilsson, C., Robertson, J. C., Rodel, R., Sindorf, N., and Wisser, D.: Global Reservoir and Dam Database, Version 1 (GRanDv1): Dams, Revision 01. Palisades, NY: NASA Socioeconomic Data and Applications Center (SEDAC). http://dx.doi.org/10.7927/H4N877QK, 2011.**

This paper discussed the regional flow regimes without the use of any hydrographs. The authors do have the daily and monthly flow records for each basin. It is very useful to show the mean hydrograph for the regions in discussion. With the hydrographs, monthly flow data will be used and presented to show the seasonality of flow and also the difference of mean flows between the regulated and unregulated basins. I would also suggest, for the regulated basins, to work out and compare the mean flows for the pre- and post-dam periods, so as to demonstrate the magnitude of dam effects on flow conditions.

**The results presented in this study are based on the constructed, continuous time series of observed discharge for 42 rivers of northern Canada. Given the availability of the daily river discharge, climatological hydrographs for each of the six regions of interest can easily be constructed and are added to a new subsection (4.4) in the Results section. A brief description of the methodology used to construct the hydrographs of daily river discharge is also now incorporated at the end of Section 3.2.2.**

**Please note however that we did not include climatological hydrographs for individual, regulated/unregulated rivers (pre-/post-regulation) as suggested by this referee as some of the hydrometric data needed for these remain proprietary by hydropower companies that have not provided their consent of the publication of these results. Instead we will consider a subsequent effort in which more details of the climatological hydrographs, namely for**

**individual rivers, will be presented. Additional text has been added to the second paragraph of the Conclusions to suggest this an avenue for future research.**

Citation of papers: Rivers are very different from one to the other. Many good papers have been published in the last 10 years on northern river hydrology; some of them are extremely useful for this work over northern Canada. The authors should read and cite more papers (a short list below), particularly those on basin-scale analyses of hydrological regimes and changes induced by climate variation and human impacts.

Woo, M.K., Thorne, R., 2003: Streamflow in the Mackenzie Basin, Canada. Arctic 56, 328-340.

Peters, D.L., Prowse, T.D., 2001: Regulation effects on the lower Peace River, Canada. Hydrol. Process. 15, 3181-3194.

Yang, D., X. Shi, P. Marsh, 2014: Variability and extreme of Mackenzie River daily discharge during, 1973-2011, Quaternary International, doi/10.1016 /j.quaint.2014.09.023

Wang S. J. Huang, D. Yang, G. Pavlic, and J. Li, 2014: An assessment of long-term water budget closures for large drainage basins in Canada, Hydrological Processes, DOI: 10.1002/hyp.10343.

**As commented by Referee #1, there have been some important, recent studies on pan-Arctic river discharge with some focusing on northern Canada. While our literature survey is already quite extensive (73 references in the Discussion Paper), we have added the above references in our revised paper as they provide further context and report some of the latest findings on streamflow trends and variability observed in northern Canada. Thus the following sentence has been added to the third paragraph of the Introduction:**

**"This effort also places into context previous studies focused on hydrological variability and trends in northern Canada (e.g., Peters and Prowse, 2001; Woo and Thorne, 2003; Wang et al., 2015; Yang et al., 2015)."**

**A few other recent publications have also been added in our review of the current knowledge of observed and projected pan-Arctic river discharge, as listed below:**

**New References:**

**Tananaev, N. I., Makarieva, O. M., and Lebedeva, L. S.: Trends in annual and extreme flows in the Lena River basin, Northern Eurasia, Geophys. Res. Lett., 43, doi: 10.1002/ 2016GL070796, 2016.**

**van Vliet, M. T. H., Franssen, W. H. P., Yearsley, J. R., Ludwig, F., Haddeland, I., Lettenmaier, D. P., and Kabat, P.: Global river discharge and water temperature under climate change, Glob. Planet. Change, 23, 450-464, 2013.**

Conclusion: The paper states. . .” Of note is the augmented variability in both regulated and unregulated rivers during the most recent decade of study (2004-2013), which may be climate-driven. It is clear from many other studies that the regulated basins are not reliable to reflect climate change impact to basin hydrology change. The argument in this paper about climate-driven similar changes in flow variation in regulated and unregulated basins might be true; however, the fact remains that with the strong dam regulation, it is difficult or impossible to detect climate signal over a basin. Thus, it is always better to rely on flow data and info from the unregulated rivers.

**We agree with Referee #1 that it is difficult or nearly impossible to detect climate change signals for regulated rivers. However, on p. 20 lines 11-18 we outline cases where enhanced variability in the most recent decade in seen in both paired nearby regulated and unregulated rivers. Therefore, we feel comfortable with the soft statement "augmented variability....may be climate-driven".**

Finally, at the end of the Conclusion Section, the paper states "Use of a hydrological model forced by output from global climate models under various scenarios also allows projections of future discharge across northern Canada. This work is currently being undertaken by the authors with the Arctic-HYPE hydrological model (Andersson et al., 2015) for the Hudson and James Bay drainage basin." This work is interesting, but it is unclear if the ongoing modeling work has considered human activities, particularly dams in the basins.

**This modelling work will consider the effects of regulation on streamflow. We can clarify this ongoing effort with the following modification to the conclusion:**

**"Use of a hydrological model forced by output from global climate models under various scenarios also allows projections of future discharge across northern Canada. This work is currently being undertaken by the authors with the Arctic-HYPE hydrological model (Andersson et al., 2015) for the Hudson and James Bay drainage basin under existing flow regulation practices."**

Referee #2:

General Comments:

This manuscript, titled "Recent trends and variability in river discharge across northern Canada", summarizes interdecadal trends and interannual variability in river discharge for 42 principal rivers draining northern Canada over a 50-year period. The research focused on the science question "whether or not river discharge in northern Canada shows a continued decrease in the twenty-first century as first reported in the leading author's previous publication". Also, the

effects of flow regulation and climate variability are both considered in the manuscript. The leading author has much experience on the similar topic. Also, the manuscript is clearly written. Therefore, the reviewer would recommend this manuscript for publication.

Moreover, the reviewer would suggest the following specific comments to improve the clarity of the paper.

**Sincere thanks for this general overview and positive assessment of our Discussion Paper. Please consult our responses below to the three specific comments provided by Referee #2.**

Specific Comments:

1) In northern Canada, there are substantial gaps in the time series of river discharge. The authors used a two-step process similar to Déry et al. (2005a). However, the streamflow timing change was not considered in this gap filling strategy. As indicated in the manuscript, the gap-filling process can influence the magnitude of MKT trends. Therefore, the uncertainty resulting from streamflow timing change should be included in Section 5 (Discussion).

   **The referee is correct in pointing out that the two-step strategy used to fill in gaps in the river discharge time series can be influenced by streamflow timing changes. As such, a precautionary note is now included in section 3.2.2 (Statistical and Trend Analyses) where impacts of the gap-filling strategy on trend and statistical analyses is provided. The text is added here rather than in Section 5 (the Discussion) as it complements the discussion of some of the caveats of the approach to in-fill missing streamflow data already included in the paper. The added text reads as:**

   **"Additional uncertainty in the trend analyses arises from potential shifts in the timing of streamflow that may otherwise be missed by the gap-filling process; however, examination of results based on annual and seasonal river discharge data attenuates this issue."**

2) For some river basins affected by seasonally frozen soils and permafrost in northern Canada, there are significantly increased groundwater to river discharge due to the thawing of seasonally frozen soils and permafrost. For their contribution in trends and variability of river discharge, the authors mentioned in the text but without further discussion. Also, the related references are from 1986 (Woo) to 2007 (Walwoord and Striegl), which need to be updated and be consistent with your analysis period.

   **Permafrost degradation and thawing is one process further altering hydrological processes in Canada's cold environments. While not the focus of the present study,**

**it is important to remind the reader of the potential influence of changes in seasonally frozen ground and permafrost distribution on streamflow generation processes in northern Canada. To that end, the recent study by St. Jacques and Sauchyn (2009) is added to the text in the Discussion (Section 5.1) as it provides evidence of an increasing influence of permafrost thawing on winter baseflow observed in northern Canada's rivers. Note however that we are not aware of a comprehensive study exploring the role of changes in seasonally-frozen ground and permafrost degradation on river discharge across northern Canada matching exactly our study period of 1964-2013.**

**"St. Jacques and Sauchyn (2009) likewise reported significant increases in winter (base) flows for the Mackenzie and other rivers of the Northwest Territories in possible response to permafrost degradation.**

**New Reference:**

**St. Jacques, J.-M. and Sauchyn, D. J.: Increasing winter baseflow and mean annual streamflow from possible permafrost thawing in the Northwest Territories, Canada, Geophys. Res. Lett., 36, L01401, doi: 10.1029/2008GL035822, 2009.**

3) Although the effects of flow regulation and climate variability were both discussed in the manuscript, their effects were not split clearly in the analyses. It would be helpful to use the modelling tool and separate their effects as part of your future work. Please discuss this in Section 5.

**Indeed both climate change and anthropogenic developments have affected and will continue to influence river discharge in northern Canada. Thus we have added a sentence at the end of Section 5.2 (Anthropogenic influences) invoking the use of modelling tools to detect hydrological changes from both climate change and flow regulation, as follows:**

**"Thus the use of land surface or hydrological models is particularly useful in identifying the individual roles of climate change and anthropogenic activities on streamflow variability and trends that otherwise may be masked in the observational data."**